# The impact of missing data rates and imputation methods on the assumption of unidimensionality

Ayman Omar Baniamer ⓘ *

Curriculum and Teaching Department, The World Islamic Sciences and Education University W.I.S.E, Amman, Jordan

* ayman.baniamer@wise.edu.jo

## Abstract

Statistical models are essential tools in data analysis. However, missing data plays a pivotal role in impacting the assumptions and effectiveness of statistical models, especially when there is a significant amount of missing data. This study addresses one of the core assumptions supporting many statistical models, the assumption of unidimensionality. It examines the impact of missing data rates and imputation methods on fulfilling this assumption. The study employs three imputation methods: Corrected Item Mean, multiple imputation, and expectation maximization, assessing their performance across nineteen levels of missing data rates, and examining their impact on the assumption of unidimensionality using several indicators (Cronbach's alpha, corrected correlation coefficients, factor analysis (Eigenvalues ($\lambda_1$, $\lambda_1/\lambda_2$, and $\frac{\lambda_1 - \lambda_2}{\lambda_2 - \lambda_3}$), cumulative variance, and communalities). The study concluded that all imputation methods used effectively provided data that maintained the unidimensionality assumption, regardless of missing data rates. Additionally, it was found that most of the unidimensionality indicators increased in value as missing data rates rose.

## Introduction

Data preparation is the starting point for good data analysis. This process involves paying attention to the existence of missing data, outliers, or errors that render the data unsuitable for analysis, such as coding errors, measurement level errors (nominal, ordinal, categorical, and ratio), or determining its type (continuous, discrete, and qualitative). It also includes considering whether the data is part of a time series or has multiple dimensions. All of this must be considered before data analysis to identify the appropriate methods for handling data and to ensure the quality of the final results of the analysis.

Unidimensionality refers to the property of a set of observed variables (or test items) that are assumed to measure a single latent trait. Formally, a test is considered unidimensional if the covariance among its items can be adequately explained by one common factor. Mathematically, if $X_i$ represents the score on item i (for

**Data availability statement:** All relevant data for this study are publicly available from the figshare repository (https://doi.org/10.6084/m9.figshare.28354868).

**Funding:** The author(s) received no specific funding for this work.

**Competing interests:** The authors have declared that no competing interests exist.

i = 1,…,k), then under the unidimensional model, the observed data matrix X can be expressed as: $X = \mu + \lambda F + \epsilon$, where μ is the vector of item means, λ is a vector of factor loadings, F is the common latent factor (or trait), and $\epsilon$ represents unique variances specific to each item. This model implies that a single factor is sufficient to account for the covariation among items, with any remaining variance attributed to item-specific measurement error or unique factors [1].

The assumption of unidimensionality plays a fundamental role in many models used in psychological and educational analysis, as it helps ensure the test items or scales measure only one trait or ability, thereby facilitating the interpretation of results and enhancing the reliability of measurement tools [2]. In the Item Response Theory (IRT) for instance, the assumption of unidimensionality ensures that performance variations among individuals are attributed to differences in their abilities or personal traits, which in turn enhances the process of assessing their capabilities [3]. The Rasch model heavily relies on this assumption to ensure the consistency of estimates, thereby achieving fair and reliable results [4]. The assumption of Unidimensionality also makes it possible to link an individual's responses to a factor or a single trait, allowing for the ability to compare individuals based on that trait, and aiding in data interpretation [5]. When this assumption is violated, the interpretation of scores becomes more complex, and the relationship between items and ability becomes less clear, which weakens the predictive power [6]. Therefore, verifying Unidimensionality is essential when constructing new tests, as it ensures the test items work together consistently toward measuring the specified target, contributing to the improvement of test quality [5,7].

The assumption of Unidimensionality is the basis for many statistical models used in psychological and educational analysis. Many statistical models assume Unidimensionality to be effective and accurate. The classical test theory is based on the idea that each item in the test measures the same dimension. The Rasch Model is one of the most common models that rely on this assumption, as it assumes that all items measure one trait, which allows for accurate and consistent estimation of ability across all individuals [8]. There are also logistic IRT models (1PL, 2PL, and 3PL). The Guttman Scale also assumes that individuals who answer an item correctly will respond to all previous items correctly, so he arranged the individuals in a graded sequence according to the essential trait that items measure [2].

The Unidimensionality assumption can be verified by using several indicators. Several of these indicators and the criterion for judging each indicator, such as the H index (Hattie's Unidimensionality index) which measures the extent to which the test items relate to one basic dimension, as well as indicators based on Eigenvalues or Residuals, He also mentioned a group of Indices Based on Factor Analysis, and Indices Based on Latent Trait Models [5]. A previous study mentioned the Omega Hierarchical index (Ωh), which measures the proportion of total variance explained by general factors [9]. Moreover, they mentioned an index based on the division of common variance in data, into single components and multiple components, which is the explained common variance index, indicates that most of the variance is common and explained by one dimension [10]. The provided Essential Unidimensionality Index assesses how close data are too ideal Unidimensionality [11].

A study found that few researchers consider the impact of missing values on their models, often treating them as a minor issue or nuisance to be ignored [12]. However, missing data is a common problem in scientific research and statistical analysis, as it significantly affects the decisions made about the results of data analysis. Missing data impacts the power of statistical tests and may lead to a bias in parameter estimating [13], it also affects the statistical analysis techniques used, as most statistical procedures require a response for each variable or item [14, 15]. Best practice guidelines recommend that every quantitative study should report the extent and nature of missing values, as well as the rationale and procedures used to handle them [16]. The reasons for missing data may vary, sometimes due to respondent's refusal to answer certain elements, failure to answer them, inadequate response time, errors in data entry, or the limitation of attrition [13].

## Missing patterns and mechanisms

Missing data may follow several patterns, it may take the arbitrary pattern, univariate Pattern when missing occurs with only one variable, multivariate pattern When there is more than one variable containing missing data [13], or monotone pattern when data missing takes graded form in some independent variables at all time intervals, rather than at a specific time point, so that the first variables have the highest data missing, followed by the variable with the second highest missing, and so on [17]. In contrast, missing may take the Non-Montone Pattern, or so-called General Pattern, when the missing does not take the graded shape, as the missing may take the File Matching Pattern, when missing occurs in two independent variables, and there is no common missing data between the two variables [18]. Missing patterns in data may be classified as "connected" or "unconnected" depending on the relationship between the missing of one observation and the missing of other observations across a horizontal or vertical series of data.

Rubin [19] proposed three mechanisms of missing data, The first mechanism is Missing Completely At Random (MCAR), which means that the probability of missing data is the same for all cases, indicating that the causes of missing are not related to the data itself, to any of the study variables or the circumstances of the study experiment, and this enables us to ignore many of the complexities that can arise because of the missing data, and although this scenario is convenient, it may be unrealistic. But If the possibility of missing data is only within a specific set of observed data, or is directly or indirectly linked to external variables; That is the missing data is not influenced by the characteristics of individuals or by the characteristics of the items themselves, but influenced by the characteristics of other variables, it will be the Missing At Random (MAR), which is a much broader category of MCAR, and more general and realistic, as modern methods of handling missing data usually start from the MAR assumption. If the two previous cases do not apply, we are referring to Missing Not at Random MNAR, and sometimes that means that missing occurred for unknown reasons, which is the most complex case, so to deal with this type of missing we need to find more information about the causes of missing. This study used MCAR to make missing data, to provide a control in which the missingness is unrelated to any observed or unobserved data. This assumption allows for isolating and directly assessing the impact of imputation methods on the unidimensionality without the confounding effects that may arise from MAR or MNAR mechanisms.

## Imputation methods

There are several methods for completing data, rather than deleting incomplete data, such as imputing missing data; these are the methods based on replacing missing values with estimated values, and these methods usually use the original data as a basis for imputing missing values. Imputation methods are characterized by their ability to restore the data matrix while maintaining balance, meaning the same number of individuals in each experimental case. Rubin [15] and Little & Rubin, [13] were among the first to propose comprehensive treatments for this topic, however, extreme caution must be exercised when using imputation methods. Imputation methods are attractive and dangerous at once, they are attractive because they may tempt the researcher to believe that the data are complete, and they are dangerous because they combine several cases in which problems are simple enough to be dealt with legitimately, and cases in which standard

estimators applied to the real and imputation data contain significant biases [20]. No method can be preferred over others to impute missing values. A study conducted to compare six methods for handling missing data showed there is no optimal method for dealing with missing values in general [21].

Imputation methods for handling missing data can be categorized into several types based on their principles and assumptions, such as, simple methods, regression methods, Machine Learning methods, Probabilistic and Bayesian Methods, Matrix Factorization-Based Imputation, Deep Learning-Based Imputation, Hybrid and Ensemble Imputation Methods, and Domain-Specific Imputation Techniques, A new set of imputation methods has recently emerged, utilizing Generative Adversarial Networks (GANs) and Variational Autoencoder (VAE)-based imputation techniques. The GANs introduced by Goodfellow et al. [22], have gained significant attention as a deep learning-based method for imputing missing data. GANs consist of a generator, which creates synthetic data, and a discriminator, which evaluates how realistic the generated data is.

Variational Autoencoders (VAEs) were introduced by Kingma & Welling [23] as a probabilistic generative model that learns latent representations of data. Unlike traditional autoencoders, VAEs impose a probabilistic structure on the latent space, effectively handling missing data imputation by generating realistic data points that follow the learned distribution [24]. f-divergence functions, namely cross-entropy, Kullback-Leibler (KL), reverse KL, and Jensen-Shannon were identified by [25], can be effectively integrated with the generative adversarial imputation network to generate imputed values without any assumptions, and mathematically prove that the distribution of imputed data using sc-fGAIN algorithm is same as the distribution of original data. An attempt to overcome the challenges of imputation methods, a model based on GANs was proposed by [26], this ensures the generalizability of the model and the reasonableness of the complementation results, through conducting experiments on three large-scale datasets and compare them with traditional complementation methods. The experimental results show that imputeGAN outperforms traditional complementation methods in terms of accuracy of complementation. Tashiro et al. [27], propose a Conditional Score-based Diffusion model for Imputation (CSDI), a novel time series imputation method that utilizes score-based diffusion models conditioned on observed data, which is explicitly trained for imputation and can exploit correlations between observed values. It shows that, on healthcare and environmental data, CSDI improves by 40–65% over existing probabilistic imputation methods on popular performance metrics. In addition, deterministic imputation by CSDI reduces the error by 5–20% compared to the state-of-the-art deterministic imputation methods.

In this study, the researcher examined three methods for imputing missing data, MI, EM, and CIM, these three methods were selected because they represent distinct theoretical approaches to handling missing data: a simple mean-based correction, a likelihood-based iterative approach, and a multiple imputation framework that accounts for imputation uncertainty, and for their Simplicity, Practicality, and Robustness Under MCAR. This diverse selection allows a comprehensive investigation of how different theoretical underpinnings and computational strategies affect the preservation of unidimensionality in measurement models.

**Multiple Imputation (MI).** Since Rubin presented an important and influential approach to the statistical analysis of incomplete data and has been revised several times [17,28–30].

Its applications have expanded to include many fields, and it has been integrated into several statistical packages. The main idea is to replace each missing value with a set of possible values, and each value is a Bayesian draw from the conditional distribution of the missing observation given the observed data, in such a way that the set of replacements correctly represents the information about the missing value in the original data for the chosen model. The MI method does not attempt to estimate each missing value through generated values but rather aims to represent a random sample of missing values, and this process leads to valid statistical conclusions that adequately reflect the uncertainty resulting from missing values. The imputation process includes three phases:

(1) Missing values are filled several (m) times, to generate an (m) number of completed datasets.

(2) The generated complete datasets are analyzed using standard procedures.

(3) Results from completed datasets are combined to conclude.

## Expectation Maximization Algorithm (EM)

It is a multi-purpose iterative algorithm, used to calculate maximum likelihood estimates in parametric models for incomplete data. Each process includes a repetition of two steps: The expectation step (E step) and the maximization (M step). The EM algorithm was formulated and provided a general and unified formulation of an algorithm, explained its fundamental properties, and presented numerous examples and applications of it [20].

The EM algorithm revolutionized the analysis of incomplete data, making it possible to efficiently compute parameter estimates, thereby eliminating the need for random methods such as case deletion in many statistical problems [13,17,31].

The fundamental idea behind the EM algorithm is to link the incomplete data set to a set of completed data, in which the calculation of maximum probability estimates is more computationally easy to tractable. The algorithm begins with suitable initial values for the parameters, then the expectation and maximization steps are repeated until convergence is reached. By providing a set of parameter estimates, such as the vector mean and covariance matrix in the case of multivariate normality, the Expectation step calculates the conditional expectation of the log-likelihood of the complete data given the observed data and parameter estimates. This step is often reduced to calculate simple sufficient statistics. After calculating the log-likelihood of the complete data, the maximization step then finds the parameter estimates that maximize the log-likelihood of the complete data from the expectation step. The MI and EM imputation methods implemented in SPSS v.25 are used to impute missing values.

## Corrected Item Mean (CIM)

It is a technique aimed at addressing missing values through item-level and individual-level information, with the correction of imputation values based on the individual's ability. This ensures that the imputed value reflects the general trend of responses at the item level and the respondent's ability compared to others. What distinguishes this method from others is its reliance on multidimensionality in the data and individual differences in ability, making it more accurate and less biased. The first to introduce the CIM method as an improvement over previous imputation methods was Huisman [32], followed by [33] who used it in the context of psychometric data, and then [34], expanded the applicability of this method, particularly in the context of multidimensional data. Additionally, a recent study employed it in experimental studies and simulation studies, especially when there was randomly missing data [35]. However, the researcher used the formula provided by [32] in this study:

$$x_{ij} = \left( \frac{\sum_{j \in obs(i)} x_{ij}}{\sum_{j \in obs(i)} \overline{x_{\cdot j}}} \right) \overline{x_{\cdot j}} = \left( \frac{PM_i}{\frac{1}{\#obs(i)} \sum_{j \in obs(i)} IM_j} \right) IM_j$$

where $\overline{x_{\cdot j}}$ is the mean across all observed scores on item j, obs(i) is the set of all observed scores for respondent i, and $\#obs(i)$ is the number of observed scores for respondent i; $PM_i$ is the person mean for respondent i, and $IM_j$ is the item mean for item j.

The issue of missing data resulting from applying various measurement tools is a concern for researchers, as it is rare for researchers to apply measurement tools to assess a specific trait without encountering some missing data in participants' responses. Ignoring this missing data affects the quality and strength of the statistical indices and may violate some assumptions of the models used, which raises doubt about the models used. Many studies compared several imputation methods in several fields, but based on the researcher's knowledge, no study compared the effect of these methods on the assumption of unidimensionality, the assumption of unidimensionality underlies many psychometric models, ensuring that a single latent trait explains the observed item responses. When missing data occurs, improper imputation techniques can distort factor structures, leading to biased parameter estimates and misinterpretations of the construct being measured.

A robust theoretical foundation should discuss how various imputation methods, such as Expectation-Maximization (EM), Multiple Imputation (MI), and CIM, preserve or violate unidimensionality assumptions. Practically, researchers must assess the impact of these methods on unidimensionality indices to ensure methodological Discipline, so this study aims to examine the impact of missing data imputation methods through different missing data rates and its effect on the assumptions of unidimensionality. Therefore, the study addresses three methods for imputing missing data: MI, EM, and CIM. It will also explore how these methods interact with different missing data rates and their effect on the unidimensionality assumption.

## Method

### Data generation

Responses of (5000) examinees, on a test of (50) items, were generated by WINGEN3 software, developed by [36]. According to the graded model, with responses (1, 2, 3, 4, 5), the ability and difficulty are normally distributed, while discrimination was distributed with a mean of (1.8) and a standard deviation of (0.5), these parameters were chosen to ensure sufficient statistical power and reliable estimation of the unidimensionality assumption while keeping the computational demands reasonable. Additionally, these settings are consistent with similar simulation studies in the literature. missing data was introduced using R software, through Lava package 1.8.0, according to the missing completely at random (MCAR), with missing rates of (1%, 2%, 3%, 4%, 5%, 6%, 7%, 8%, 9%, 10%, 11%, 12%, 13%, 14%, 15%, 20%, 25%, 30%, 35%, 40%, 45%, 50%). The missing rate was varied from 1% to 50% to cover a wide range levels of missingness. 50% selected as the upper limit because missing rates above this rate are rare in practical applications and often lead to excessive uncertainty that may obscure the effects of imputation on the measurement model. This range allows exploring the robustness of each imputation method within a realistic boundary of missingness encountered in empirical research. The MCAR was tested using Little & Rubin's test [37], illustrated in Table 1.

Three missing rates (1%, 3%, and 5%) were removed, due to their failure to meet the assumption of MCAR. Nineteen missing rates were retained to extrapolate the effect of imputation methods across these rates on the assumption of unidimensionality.

### Data analysis

Missing values in data were imputed using the CIM in an Excel sheet. Then it was converted to SPSS v. 26 to round the imputed values to the nearest integer, SPSS was also used to impute the missing data using the EM with (1000) iterations, and MI, with (100) replications and (1000) iterations, The imputed values were also rounded to the nearest integer according to both methods. After that, the absolute differences between the original values and the imputed values were calculated in the three imputation methods to determine whether the imputation methods distort the true values of the data. Moreover, to explores the difference between the imputed values and the original values.

Table 2 also illustrates the average of matching and distortion ratios for data across three imputation methods and for all missing rates. If the absolute difference between the original value and the imputed value is zero, this indicates there is no distortion in the original data. However, if the difference between one and four, this indicates there is distortion in the original data caused by the imputation methods. Then, the reliability coefficients were found using Cronbach's alpha index as a preliminary indicator for unidimensionality, as illustrated in Table 3. Subsequently, the corrected correlation coefficients between the imputed values and the overall tool were calculated as an indicator of unidimensionality, by determining the effectiveness of the three imputation methods in producing values consistent with the overall tool and contributing effectively to measuring traits that the tool measures as a whole, through the different missing rates in data.

Finally, the factor analysis was conducted to extract Eigenvalues, using the principal component method and the promax rotation. Subsequently, Eigenvalues indices $\lambda_1$, $\lambda_1/\lambda_2$, and $\frac{\lambda_1-\lambda_2}{\lambda_2-\lambda_3}$ were calculated, as shown in Appendix 1, where $\lambda_1$ indicates the proportion of variance explained by the first principal component in Principal Component Analysis (PCA) or the first factor in Exploratory Factor Analysis (EFA), $\lambda_1/\lambda_2$ indicates the ratio of the first to the second eigenvalue, and

**Table 1. *Missing completely at random test.***

**Little's MCAR test**

| % | $\chi^2$ | df | sig. | Status |
|---|---|---|---|---|
| 1 | 23572.594 | 22868 | 0.001 | deleted |
| 2 | 53518.595 | 53587 | 0.582 | |
| 3 | 84220.361 | 83496 | 0.038 | deleted |
| 4 | 114717.379 | 114312 | 0.198 | |
| 5 | 146755.035 | 145654 | 0.021 | deleted |
| 6 | 172148.583 | 172329 | 0.62 | |
| 7 | 189891.198 | 189119 | 0.105 | |
| 8 | 203777.092 | 203539 | 0.354 | |
| 9 | 211638.604 | 211723 | 0.551 | |
| 10 | 216496.276 | 216537 | 0.524 | |
| 11 | 218127.281 | 217814 | 0.317 | |
| 12 | 217315.303 | 217319 | 0.502 | |
| 13 | 216313.214 | 216294 | 0.488 | |
| 14 | 214610.481 | 214605 | 0.496 | |
| 15 | 211944.765 | 211890 | 0.466 | |
| 20 | 199895.543 | 199946 | 0.531 | |
| 25 | 187227.846 | 187276 | 0.531 | |
| 30 | 175291.188 | 175339 | 0.532 | |
| 35 | 162222.502 | 162272 | 0.534 | |
| 40 | 149517.806 | 149561 | 0.531 | |
| 45 | 136904.887 | 136914 | 0.506 | |
| 50 | 125513.599 | 125580 | 0.552 | |

**Note: %**: Missing rate, χ2: chi-square, df: degree of freedom, Sig: Significant ≤ 0.05.

$\frac{\lambda_1 - \lambda_2}{\lambda_2 - \lambda_3}$ indicates the gap between the first and second eigenvalues to the gap between the second and third. Then, the cumulative variance explained by the first three components, as illustrated in Table 4, and the communalities for each test item were calculated for each imputation method and various missing rates, as illustrated in Table 5. Data generation and analysis are illustrated in Fig 1.

## Results

### Differences between the original and the imputed values

The results related to the absolute differences between the original values and the imputed values from the three imputation methods CIM, EM, and MI, along with various missing rates, indicate that the values imputed using the EM method were the most consistent with the original data, compared to the CIM and MI methods across different missing rates, with an average consistency of (67.26%). In contrast, the values imputed using the CIM method were the least consistent with the original data among the three methods, with an average consistency of (53.92%) across all missing rates. It is also noted that the matching ratios between the original values and the imputed values using the three imputation methods were not affected by different missing rates. The matching ratios were similar for each method, regardless the varying of missing rates. Table 2 illustrates the distortion ratios in the original data for each imputation method across different missing rates, as indicated under the "Distortion" column. The last column presents the actual missing ratios under the title "Observed", which were very close to the assumed ratios when the missing was conducted using the Lava package in R.

Table 2. Absolute difference between missing rates as compared with ZERO rate of missing due to imputation method.

| Adopted% | CIM | | | | | | EM | | | | | | MI | | | | | | Observed% |
|---|---|---|---|---|---|---|---|---|---|---|---|---|---|---|---|---|---|---|---|
| | 0 | 1 | 2 | 3 | 4 | DISTORTION | 0 | 1 | 2 | 3 | 4 | DISTORTION | 0 | 1 | 2 | 3 | 4 | DISTORTION | |
| 2% | 53.41 | 41.45 | 4.65 | 0.47 | 0.02 | 46.59 | 66.80 | 28.61 | 4.04 | 0.49 | 0.06 | 33.20 | 66.57 | 28.84 | 4.08 | 0.47 | 0.04 | 33.43 | 2.0304% |
| 4% | 53.91 | 41.18 | 4.53 | 0.32 | 0.06 | 46.09 | 67.95 | 27.80 | 3.71 | 0.49 | 0.05 | 32.05 | 68.03 | 27.63 | 3.81 | 0.48 | 0.05 | 31.97 | 3.9292% |
| 6% | 54.09 | 40.87 | 4.60 | 0.43 | 0.00 | 45.91 | 67.56 | 28.29 | 3.68 | 0.45 | 0.02 | 32.44 | 67.34 | 28.47 | 3.74 | 0.42 | 0.03 | 32.66 | 6.0204% |
| 7% | 54.06 | 40.96 | 4.58 | 0.39 | 0.01 | 45.94 | 67.58 | 28.36 | 3.55 | 0.48 | 0.02 | 32.42 | 67.54 | 28.29 | 3.66 | 0.48 | 0.03 | 32.46 | 6.9560% |
| 8% | 53.91 | 41.05 | 4.65 | 0.35 | 0.04 | 46.09 | 67.71 | 28.28 | 3.54 | 0.43 | 0.04 | 32.29 | 67.52 | 28.39 | 3.58 | 0.47 | 0.04 | 32.48 | 8.0296% |
| 9% | 53.80 | 41.12 | 4.63 | 0.42 | 0.02 | 46.20 | 67.91 | 28.06 | 3.57 | 0.44 | 0.02 | 32.09 | 67.88 | 28.06 | 3.58 | 0.46 | 0.02 | 32.12 | 9.0388% |
| 10% | 53.58 | 41.22 | 4.82 | 0.36 | 0.02 | 46.42 | 67.13 | 28.64 | 3.86 | 0.36 | 0.01 | 32.87 | 67.01 | 28.75 | 3.85 | 0.37 | 0.02 | 32.99 | 10.0156% |
| 11% | 54.21 | 40.76 | 4.65 | 0.38 | 0.01 | 45.79 | 67.83 | 27.91 | 3.79 | 0.46 | 0.00 | 32.17 | 67.60 | 28.15 | 3.75 | 0.49 | 0.00 | 32.40 | 10.9412% |
| 12% | 53.71 | 41.09 | 4.75 | 0.44 | 0.02 | 46.29 | 67.71 | 28.00 | 3.81 | 0.46 | 0.02 | 32.29 | 67.37 | 28.37 | 3.74 | 0.49 | 0.03 | 32.63 | 11.9440% |
| 13% | 54.32 | 40.82 | 4.48 | 0.37 | 0.01 | 45.68 | 68.01 | 27.92 | 3.67 | 0.38 | 0.02 | 31.99 | 67.70 | 28.17 | 3.71 | 0.41 | 0.01 | 32.30 | 13.0944% |
| 14% | 54.32 | 40.74 | 4.48 | 0.42 | 0.04 | 45.68 | 67.64 | 28.31 | 3.55 | 0.47 | 0.03 | 32.36 | 67.57 | 28.34 | 3.56 | 0.50 | 0.03 | 32.43 | 13.9628% |
| 15% | 54.64 | 40.39 | 4.54 | 0.40 | 0.03 | 45.36 | 67.88 | 28.03 | 3.63 | 0.42 | 0.03 | 32.12 | 67.56 | 28.30 | 3.67 | 0.45 | 0.03 | 32.44 | 15.0876% |
| 20% | 53.88 | 41.03 | 4.62 | 0.44 | 0.03 | 46.12 | 67.38 | 28.46 | 3.62 | 0.53 | 0.02 | 32.62 | 67.25 | 28.57 | 3.68 | 0.48 | 0.03 | 32.75 | 20.0216% |
| 25% | 53.97 | 41.05 | 4.54 | 0.41 | 0.03 | 46.03 | 67.48 | 28.35 | 3.70 | 0.45 | 0.02 | 32.52 | 67.29 | 28.52 | 3.69 | 0.48 | 0.03 | 32.71 | 25.0896% |
| 30% | 53.88 | 41.09 | 4.62 | 0.38 | 0.03 | 46.12 | 67.07 | 28.85 | 3.65 | 0.39 | 0.03 | 32.93 | 66.90 | 29.00 | 3.67 | 0.40 | 0.03 | 33.10 | 29.8644% |
| 35% | 53.90 | 41.05 | 4.66 | 0.38 | 0.02 | 46.10 | 66.81 | 29.06 | 3.67 | 0.44 | 0.02 | 33.19 | 66.59 | 29.22 | 3.73 | 0.43 | 0.02 | 33.41 | 35.0912% |
| 40% | 53.75 | 41.01 | 4.78 | 0.43 | 0.03 | 46.25 | 66.31 | 29.39 | 3.78 | 0.49 | 0.02 | 33.69 | 66.07 | 29.57 | 3.86 | 0.48 | 0.03 | 33.93 | 40.1756% |
| 45% | 53.62 | 41.20 | 4.74 | 0.41 | 0.03 | 46.38 | 66.19 | 29.54 | 3.79 | 0.45 | 0.03 | 33.81 | 66.07 | 29.63 | 3.81 | 0.46 | 0.03 | 33.93 | 45.2344% |
| 50% | 53.53 | 41.32 | 4.70 | 0.43 | 0.03 | 46.47 | 64.93 | 30.75 | 3.79 | 0.51 | 0.02 | 35.07 | 65.31 | 30.44 | 3.74 | 0.49 | 0.02 | 34.69 | 49.7680% |
| **MEAN** | **53.92** | **41.02** | **4.63** | **0.40** | **0.02** | **46.08** | **67.26** | **28.56** | **3.71** | **0.45** | **0.03** | **32.74** | **67.11** | **28.67** | **3.73** | **0.46** | **0.03** | **32.89** | |

## Cronbach alpha

It is one of the indices mentioned by [5], some of which were used in this study. The values of Cronbach Alpha shown in Table 3 indicate that the reliability coefficients are high for the three imputation methods, across different missing rates. The value of the Cronbach alpha index for original values was (0.991557), and when using the CIM across different missing rates, the value of Cronbach alpha increases with higher missing rates, from (0.99169) at (2%) missing rate to (0.994831) at (50%) missing rate. The same applies to the other imputation methods; when using the EM, the value of Cronbach alpha ranged between (0.991666) and (0.994064), and between (0.991663) and (0.994120) for MI method, depending on the increase in missing rates, while the average Cronbach's alpha coefficient across all missing rates when using imputation methods CIM, EM, and MI was (0.992777, 0.992535, and 0.992528) respectively. The Cronbach's alpha value for the CIM method was the highest among the other imputation values, followed by the EM and MI.

## Corrected Item-Total Correlation (CITC)

It is one of the important indices in assessing unidimensionality, in addition to being used as an indicator of internal scale reliability Appendix 1 shows CITC between the imputed values and the overall scale, to determine the effectiveness of the three imputation methods in finding values consistent with the overall scale, and to contribute effectively measuring the trait that the tool as a whole assesses, through the various missing rates in data, which are considered indicators of unidimensionality.

The CITC values show that the imputation methods used were effective in compensating for the missing data, as the correlation coefficients of these imputed values with the overall tool were equal to or higher than the correlation of the original values

**Table 3.** *Cronbach's Alpha for Imputation Methods.*

|  | CIM | EM | MI |
|---|---|---|---|
| **0** | **0.991557** | **0.991557** | **0.991557** |
| 2 | 0.991690 | 0.991666 | 0.991663 |
| 4 | 0.991813 | 0.991770 | 0.991768 |
| 6 | 0.991933 | 0.991872 | 0.991867 |
| 7 | 0.991999 | 0.991920 | 0.991918 |
| 8 | 0.992081 | 0.991985 | 0.991982 |
| 9 | 0.992125 | 0.992019 | 0.992014 |
| 10 | 0.992212 | 0.992094 | 0.992089 |
| 11 | 0.992257 | 0.992121 | 0.992116 |
| 12 | 0.992340 | 0.992196 | 0.992189 |
| 13 | 0.992364 | 0.992206 | 0.992195 |
| 14 | 0.992449 | 0.992279 | 0.992275 |
| 15 | 0.992533 | 0.992344 | 0.992337 |
| 20 | 0.992861 | 0.992628 | 0.992609 |
| 25 | 0.993161 | 0.992849 | 0.992829 |
| 30 | 0.993480 | 0.993090 | 0.993077 |
| 35 | 0.993846 | 0.993393 | 0.993369 |
| 40 | 0.994245 | 0.993716 | 0.993702 |
| 45 | 0.994549 | 0.993946 | 0.993917 |
| 50 | 0.994831 | 0.994064 | 0.994120 |
| **MEAN** | **0.992777** | **0.992535** | **0.992528** |

**Note:** CIM: Corrected item mean, EM: Expectation maximization, MI: Multiple imputation.

with the tool, regardless of the missing data rates. In fact, they were slightly higher than the correlation coefficients of the original data, especially at higher missing data rates. Generally, an increase in the corrected correlation coefficient was observed with an increase in the missing data rate. The overall CITC values ranged between (0.85) and (0.91), which are high values. The results also showed that the CITC values for the three imputation methods were close at the same missing data rate. The values imputed using the EM and MI methods were the best at providing values close to the original data in terms of their correlation with the overall tool, across different missing data rates. This is evident from the overall means of the CITC found at the end of Appendix 1.

## Eigenvalues

Recent studies rely on Eigenvalues as one of the key indices for determining the number of factors in factor analysis, Therefore, three indices based on Eigenvalues were relied upon in this study. Table 4 shows these indices used for

**Table 4. Eigenvalues for the first three components, through CIM, EM, and MI imputations methods, through different missing data ratios.** $\lambda_1$ : Eigenvalues for the first component, $\lambda_1/\lambda_2$: the ratio between the first and the second components eigenvalues, $\frac{\lambda_1-\lambda_2}{\lambda_2-\lambda_3}$: the ratio between the difference of the first and second eigenvalues and the difference of the second and third eigenvalues.

| % | Index | CIM 1 | CIM 2 | CIM 3 | EM 1 | EM 2 | EM 3 | MI 1 | MI 2 | MI 3 | % | Index | CIM 1 | CIM 2 | CIM 3 | EM 1 | EM 2 | EM 3 | MI 1 | MI 2 | MI 3 |
|---|---|---|---|---|---|---|---|---|---|---|---|---|---|---|---|---|---|---|---|---|---|
| 0% | $\lambda_1$ | 36.78 | 1.88 | 0.77 | 36.78 | 1.88 | 0.77 | 36.78 | 1.88 | 0.77 | 12% | $\lambda_1$ | 37.57 | 1.61 | 0.65 | 37.49 | 1.90 | 0.78 | 37.48 | 1.90 | 0.78 |
| | $\lambda_1/\lambda_2$ | 19.55 | | | 19.55 | | | 19.55 | | | | $\lambda_1/\lambda_2$ | 23.29 | | | 19.71 | | | 19.71 | | |
| | $\frac{\lambda_1-\lambda_2}{\lambda_2-\lambda_3}$ | 31.44 | | | 31.44 | | | 31.44 | | | | $\frac{\lambda_1-\lambda_2}{\lambda_2-\lambda_3}$ | 37.34 | | | 31.71 | | | 31.65 | | |
| 2% | $\lambda_1$ | 36.91 | 1.84 | 0.75 | 36.90 | 1.89 | 0.77 | 36.90 | 1.89 | 0.77 | 14% | $\lambda_1$ | 37.68 | 1.58 | 0.65 | 37.58 | 1.91 | 0.79 | 37.58 | 1.92 | 0.79 |
| | $\lambda_1/\lambda_2$ | 20.06 | | | 19.53 | | | 19.54 | | | | $\lambda_1/\lambda_2$ | 23.84 | | | 19.63 | | | 19.62 | | |
| | $\frac{\lambda_1-\lambda_2}{\lambda_2-\lambda_3}$ | 32.14 | | | 31.38 | | | 31.42 | | | | $\frac{\lambda_1-\lambda_2}{\lambda_2-\lambda_3}$ | 38.61 | | | 31.67 | | | 31.60 | | |
| 4% | $\lambda_1$ | 37.04 | 1.80 | 0.73 | 37.02 | 1.89 | 0.77 | 37.02 | 1.89 | 0.77 | 20% | $\lambda_1$ | 38.12 | 1.45 | 0.60 | 37.99 | 1.91 | 0.79 | 37.97 | 1.92 | 0.79 |
| | $\lambda_1/\lambda_2$ | 20.62 | | | 19.59 | | | 19.59 | | | | $\lambda_1/\lambda_2$ | 26.33 | | | 19.88 | | | 19.83 | | |
| | $\frac{\lambda_1-\lambda_2}{\lambda_2-\lambda_3}$ | 33.04 | | | 31.44 | | | 31.44 | | | | $\frac{\lambda_1-\lambda_2}{\lambda_2-\lambda_3}$ | 43.23 | | | 32.14 | | | 31.97 | | |
| 6% | $\lambda_1$ | 37.16 | 1.74 | 0.70 | 37.13 | 1.89 | 0.77 | 37.12 | 1.89 | 0.77 | 30% | $\lambda_1$ | 38.83 | 1.27 | 0.55 | 38.55 | 1.94 | 0.77 | 38.53 | 1.94 | 0.77 |
| | $\lambda_1/\lambda_2$ | 21.33 | | | 19.64 | | | 19.63 | | | | $\lambda_1/\lambda_2$ | 30.63 | | | 19.84 | | | 19.86 | | |
| | $\frac{\lambda_1-\lambda_2}{\lambda_2-\lambda_3}$ | 34.07 | | | 31.43 | | | 31.41 | | | | $\frac{\lambda_1-\lambda_2}{\lambda_2-\lambda_3}$ | 52.16 | | | 31.29 | | | 31.29 | | |
| 7% | $\lambda_1$ | 37.22 | 1.72 | 0.70 | 37.18 | 1.90 | 0.77 | 37.18 | 1.90 | 0.77 | 35% | $\lambda_1$ | 39.28 | 1.15 | 0.53 | 38.94 | 1.92 | 0.78 | 38.91 | 1.92 | 0.77 |
| | $\lambda_1/\lambda_2$ | 21.60 | | | 19.59 | | | 19.60 | | | | $\lambda_1/\lambda_2$ | 34.13 | | | 20.28 | | | 20.26 | | |
| | $\frac{\lambda_1-\lambda_2}{\lambda_2-\lambda_3}$ | 34.58 | | | 31.28 | | | 31.28 | | | | $\frac{\lambda_1-\lambda_2}{\lambda_2-\lambda_3}$ | 60.93 | | | 32.44 | | | 32.23 | | |
| 8% | $\lambda_1$ | 37.30 | 1.70 | 0.68 | 37.25 | 1.89 | 0.77 | 37.25 | 1.89 | 0.77 | 40% | $\lambda_1$ | 39.77 | 1.07 | 0.49 | 39.36 | 1.92 | 0.77 | 39.33 | 1.92 | 0.77 |
| | $\lambda_1/\lambda_2$ | 21.95 | | | 19.68 | | | 19.68 | | | | $\lambda_1/\lambda_2$ | 37.21 | | | 20.53 | | | 20.52 | | |
| | $\frac{\lambda_1-\lambda_2}{\lambda_2-\lambda_3}$ | 35.03 | | | 31.41 | | | 31.43 | | | | $\frac{\lambda_1-\lambda_2}{\lambda_2-\lambda_3}$ | 66.85 | | | 32.72 | | | 32.51 | | |
| 9% | $\lambda_1$ | 37.34 | 1.67 | 0.69 | 37.29 | 1.90 | 0.78 | 37.28 | 1.89 | 0.78 | 45% | $\lambda_1$ | 40.16 | 1.00 | 0.48 | 39.66 | 1.94 | 0.78 | 39.62 | 1.95 | 0.77 |
| | $\lambda_1/\lambda_2$ | 22.33 | | | 19.67 | | | 19.69 | | | | $\lambda_1/\lambda_2$ | 40.27 | | | 20.40 | | | 20.27 | | |
| | $\frac{\lambda_1-\lambda_2}{\lambda_2-\lambda_3}$ | 36.15 | | | 31.69 | | | 31.73 | | | | $\frac{\lambda_1-\lambda_2}{\lambda_2-\lambda_3}$ | 75.28 | | | 32.30 | | | 31.93 | | |
| 10% | $\lambda_1$ | 37.43 | 1.66 | 0.67 | 37.37 | 1.90 | 0.78 | 37.36 | 1.90 | 0.78 | 50% | $\lambda_1$ | 40.56 | 0.93 | 0.45 | 39.83 | 1.92 | 0.75 | 39.92 | 1.92 | 0.75 |
| | $\lambda_1/\lambda_2$ | 22.57 | | | 19.65 | | | 19.65 | | | | $\lambda_1/\lambda_2$ | 43.72 | | | 20.74 | | | 20.76 | | |
| | $\frac{\lambda_1-\lambda_2}{\lambda_2-\lambda_3}$ | 36.09 | | | 31.60 | | | 31.62 | | | | $\frac{\lambda_1-\lambda_2}{\lambda_2-\lambda_3}$ | 83.83 | | | 32.52 | | | 32.47 | | |
| 11% | $\lambda_1$ | 37.48 | 1.64 | 0.66 | 37.40 | 1.91 | 0.77 | 37.39 | 1.91 | 0.77 | | | | | | | | | | | |
| | $\lambda_1/\lambda_2$ | 22.83 | | | 19.60 | | | 19.62 | | | | | | | | | | | | | |
| | $\frac{\lambda_1-\lambda_2}{\lambda_2-\lambda_3}$ | 36.51 | | | 31.25 | | | 31.27 | | | | | | | | | | | | | |

assessing unidimensionality, represented by the Eigenvalues of the first three components, $\lambda_1$, $\lambda_1/\lambda_2$ and $\frac{\lambda_1-\lambda_2}{\lambda_2-\lambda_3}$ through various imputation methods and multiple missing ratios. Where the corresponding values for a missing rate of 0% return to the original complete values. The results showed the Eigenvalues for the first component when using the CIM imputation method were higher than Eigenvalues when using the EM and MI methods across different missing rates. It is noted the Eigenvalues when using CIM method increase with the increase in the data missing ratio, reaching (36.91) at 2% missing rate and rising to (40.56) at 50% missing rate. Similarly, the second index $\lambda_1/\lambda_2$, it was (20.06) at 2% missing rate, increasing to (43.72) at 50% missing rate.

As for the third index $\frac{\lambda_1-\lambda_2}{\lambda_2-\lambda_3}$, it was increasing with the increase in the missing percentage, except at a missing percentage of 10%. It was (32.14) at a missing percentage of 2% and continued to reach (36.15) at a missing rate of 9%, then it declined to (36.09) at 10%, before rising again to reach (83.83) at missing rate of 50%. Additionally, the Eigenvalues when using the EM also increased with the rising missing rates, reaching (36.90) at 2%, and reaching (39.83) at missing rate of 50%. As for $\lambda_1/\lambda_2$ and $\frac{\lambda_1-\lambda_2}{\lambda_2-\lambda_3}$ the values did not follow consistent pattern with the increase in the rate of data missing; they sometimes rise and sometimes fall. When calculating unidimensionality indicators using MI, the Eigenvalues followed the same pattern observed with the CIM and EM methods, increasing with higher data missing rates. However, all of them are sufficient to assess the validity of unidimensionality assumption.

Generally, Eigenvalues indices used in this study confirm the unidimensionality assumption of data across different imputation methods (CIM, EM, and MI) and varying rates of data missing. All of them are close to the values of complete data shown in Table 4 at missing rate 0%. Additionally, the values of eigenvalues indices were higher when using CIM compared to EM and MI methods. Furthermore, the Eigenvalues indicator $\lambda_1$ showed higher values when using EM compared to MI method, except at missing rate of 50%, As for unidimensionality indicators $\lambda_1/\lambda_2$ and $\frac{\lambda_1-\lambda_2}{\lambda_2-\lambda_3}$, their values were similar between the EM and MI at different missing rates, with slight difference, reaching a maximum of (0.37) at missing rate of 45%. Appendix 2 shows the Eigenvalues of the first three components in detail for each imputation method.

## Cumulative total variance for components

One of the indicators used to assess unidimensionality is based on the percentage of variance, explained by extracted components. Usually, if the first component explains a high percentage of variance, this is considered strong evidence of unidimensionality. Table 5 shows the cumulative explained variance for the first two components according to the imputation method and missing rates. It appears the explained variance on the first component, or the cumulative variance on the first two components, based on the indicators for assessing unidimensionality through explained variance, is sufficient to determine the unidimensionality of data according to the three imputation methods, regardless of the missing rates, this means that the imputation methods CIM, EM, and MI, were effective in maintaining the assumption of unidimensionality of data, regardless of missing rates. The explained variance increased on the first component with higher missing rates; the lowest explained variance according to the three imputation methods was at missing rate of 2% at a value of (73.79), while the highest was at missing rate of 50% at a value of (81.13). Additionally, the explained variance from the first component was the highest when using the CIM, followed by the EM, and then MI. This is evident from the total mean of the explained variance for the first component across missing rates in Table 5.

## Communalities

Communalities are considered important indicators in assessing the extent to which items are related to the underlying structure of scale. The higher the commonalities, the more it supports the assumption of unidimensionality. Communalities values shown in Appendix 3 indicate appropriate Communalities for the first factor across all imputation methods, despite the varying missing rates in data. These values are close to the Communalities values of the original data at 0% missing rate. Some cases showed commonalities values exceeding those of complete data, while others showed values lower than those of complete data. Overall, the end of the Table presents the mean of Communalities for all items according

**Table 5. *Cumulative total variance for components of imputation methods.***

| % | CIM | | EM | | MI | |
|---|---|---|---|---|---|---|
| | 1 | 2 | 1 | 2 | 1 | 2 |
| 0% | 73.56 | 77.32 | 73.56 | 77.32 | 73.56 | 77.32 |
| 2% | 73.82 | 77.50 | 73.80 | 77.58 | 73.79 | 77.57 |
| 4% | 74.07 | 77.66 | 74.04 | 77.82 | 74.03 | 77.81 |
| 6% | 74.31 | 77.80 | 74.26 | 78.04 | 74.25 | 78.03 |
| 7% | 74.44 | 77.89 | 74.37 | 78.16 | 74.36 | 78.16 |
| 8% | 74.59 | 77.99 | 74.50 | 78.28 | 74.49 | 78.27 |
| 9% | 74.69 | 78.03 | 74.57 | 78.37 | 74.56 | 78.35 |
| 10% | 74.86 | 78.18 | 74.74 | 78.54 | 74.72 | 78.53 |
| 11% | 74.95 | 78.24 | 74.80 | 78.61 | 74.78 | 78.59 |
| 12% | 75.14 | 78.37 | 74.99 | 78.79 | 74.97 | 78.77 |
| 13% | 75.18 | 78.37 | 75.00 | 78.83 | 74.97 | 78.80 |
| 14% | 75.36 | 78.52 | 75.16 | 78.99 | 75.15 | 78.98 |
| 15% | 75.56 | 78.66 | 75.33 | 79.14 | 75.31 | 79.12 |
| 20% | 76.24 | 79.14 | 75.98 | 79.81 | 75.94 | 79.77 |
| 25% | 76.90 | 79.61 | 76.50 | 80.35 | 76.45 | 80.31 |
| 30% | 77.67 | 80.20 | 77.10 | 80.99 | 77.06 | 80.94 |
| 35% | 78.56 | 80.86 | 77.88 | 81.73 | 77.81 | 81.65 |
| 40% | 79.53 | 81.67 | 78.71 | 82.55 | 78.66 | 82.50 |
| 45% | 80.33 | | 79.33 | 83.22 | 79.23 | 83.14 |
| 50% | 81.13 | | 79.65 | 83.49 | 79.83 | 83.68 |
| MEAN | 76.18 | 78.75 | 75.83 | 79.65 | 75.81 | 79.63 |

to the imputation method and missing rates, which demonstrate that communalities were generally higher than those in complete data in most results, and Communalities increase with the rise in missing rates. Communalities were also close to each other for the imputation methods at the same missing rate, especially when using the EM and MI, where Communalities were equal in most cases. As shown in Appendix 3, some Communalities were below 0.50 for the original data, due to data generation assumptions.

## Discussion

This study highlighted that the three imputation methods MI, EM, and CIM effectively imputed missing data, closely matching the original data. Across different missing data rates, the consistency in matching across missing data levels suggests that the EM method may be more suited to ordinal data, as noted by [34], who identified EM as one of the best methods for Likert data. Additionally, CIM may serve as a simpler alternative to more complex imputation methods for categorical missing data. This finding aligns with Xu et al. [38], who compared four imputation methods (direct deletion, mode imputation, hot-deck (HD) imputation, and MI) and found MI to be the least biased across missing data rates. The EM and MI methods are ideal for imputing missing values in various contexts [39].

Regarding the assumption of unidimensionality, overall metrics indicated that unidimensionality was preserved after imputing missing data using MI, EM, and CIM, regardless of missing data rates. Cronbach's alpha was high across all three methods and close to the values from the original data, increasing with higher levels of missing data. High-reliability coefficients generally suggest unidimensionality, as confirmed by [40–42], they suggested that values between (0.70) and (0.90) are acceptable, and values above (0.90) indicate strong similarity among items, preliminarily indicating unidimensionality. In a previous study, they found that unidimensionality cannot be assumed solely based on Cronbach's alpha,

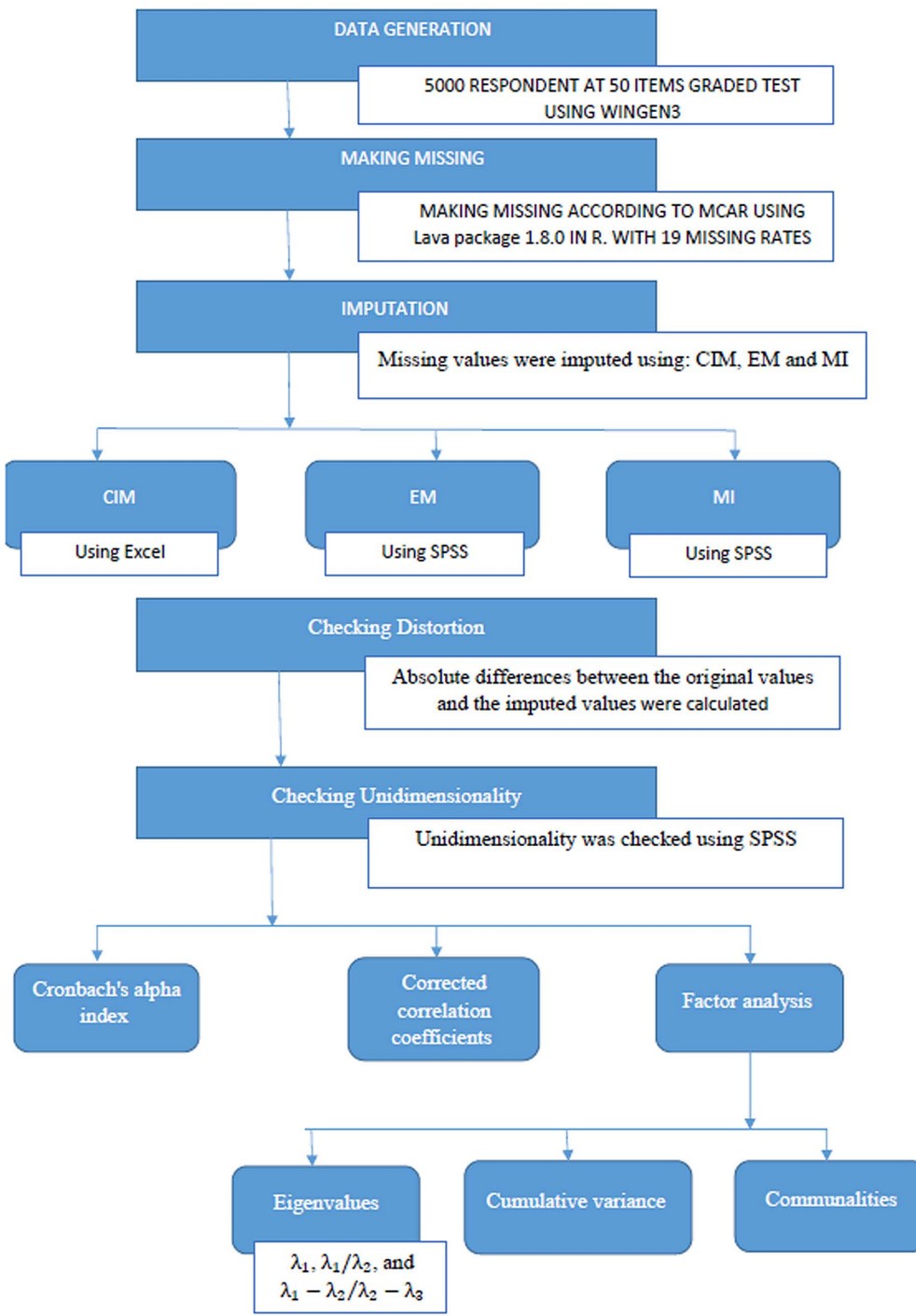

**Fig 1. Flowchart for the data generation and analysis method.**

recommending exploratory factor analysis for validation [43]. Another study found similar emphasized the need for proper interpretation and education regarding Cronbach's alpha [44]. The researcher highlights limitations in accurately assessing measurement reliability, especially for multidimensional scales, suggesting the omega coefficient and the GLB indicator as alternatives [45]. Regarding the CITC values increases were observed with higher rates of missing data. The MI and EM performed slightly better compared to CIM. The higher CITC value suggests a unidimensional structure [41,45,46]. While in others they specified that CITC values should exceed (0.30) to ensure item consistency with the overall scale [42,44]. To confirm unidimensionality, EFA and CFA are recommended.

Additional indicators are used to assess unidimensionality, with Eigenvalues showing that these indicators were highest for the CIM method compared to MI and EM methods, and values generally increased with higher missing data rates. Using Eigenvalues with parallel analysis provides more accurate factor determination and recommends additional indicators such as the Scree plot and AIC and BIC indices for further validation of the factor structure [47]. Eigenvalues greater than one may be a useful criterion for determining the number of factors, though they may be inaccurate in multi-dimensional datasets [48]. Warned solely on Eigenvalues > 1 can lead to misestimating the number of dimensions, hence recommending the use of parallel analysis and the Scree plot for greater precision [49]. Our findings are consistent with [50], which supported this approach, while Goretzko [51], found that solely relying on Eigenvalues could lead to overestimating the number of dimensions. Another study indicated that if the $\lambda_1/\lambda_2$ ratio exceeds 3–5, this supports the assumption of a dominant single factor [52]. Moreover, if the $(\lambda 1-\lambda 2)/(\lambda 2-\lambda 3)$ value is greater than 2 or 3, this further supports unidimensionality. Based on these three Eigenvalue indices, the data supports the unidimensionality, which confirms the effectiveness of the three imputation methods in preserving the factor structure of the data.

The results also showed that the Cumulative Total Variance mean for Components, after imputing missing data using the three methods, increased with higher missing data rates, with generally high values indicating the assumption of uni-dimensionality. A large proportion of variance explained by the first component is often sufficient to assume unidimensionality, a perspective supported by [7,53], suggested that the first factor should explain at least 20% of the variance, while Carmines and Zeller [54] posited that the first component should explain 40% of the total variance to affirm that items measure a single dimension. Another study recommended that the first component's explained variance should be 60% or higher to support unidimensionality [52], whereas Henson and Roberts [49] proposed a range of 50–60%. Most studies concur that cumulative explained variance alone should not be the sole indicator of unidimensionality.

To further evaluate unidimensionality, an additional indicator was used, focusing on Communalities. The results showed appropriate communality values for judging unidimensionality, with similar values across the three imputation methods at the same missing data rates. Communality values were equal to, close to, or sometimes higher or lower than those in the original data, attributable to the assumptions in the generated data distribution. Most studies indicate that commonalities above (0.50) strongly suggest unidimensionality, and our results confirm previous studies [47,55,56]. Other studies suggested that commonalities between (0.50) and (0.60) indicate unidimensionality [55], while others asserted that values above (0.60) support this assumption [52]. The researcher concluded that commonalities between (0.40) and (0.70) are strong indicators of unidimensionality, whereas values below (0.30) suggest that items might not align well with the main factor. A study conducted to examine the effect of imputation methods on factor analysis with missing data, compared to complete data factor analysis, found that EM captured the complete data factor loadings more accurately than other methods when comparing 12 imputation techniques [57].

Based on all the indicators used in this study, results showed that the three imputation methods CIM, EM, and MI effectively imputed missing values, while maintaining the assumption of unidimensionality for the data. No effect of different missing data rates on the effectiveness of the imputation methods was observed. This suggests that these methods can effectively recover missing values without significantly distorting the data's underlying factor structure. Furthermore, the performance of these methods remained stable across different missing data rates, reinforcing their robustness in handling missingness in psychometric applications. These results conform to prior studies investigating the impact of missing data imputation on factor

analysis and measurement reliability. For instance, MI and EM methods tend to produce unbiased parameter estimates when the missing data mechanism is either (MCAR) or (MAR) [58]. Similarly, CIM despite its simplicity, can maintain structural validity when missing data rates are moderate [34]. Findings corroborate these studies by showing that all three methods are Practical for preserving the assumption of unidimensionality, which is essential for accurate measurement in psychological and educational assessments. However, the study also underscores the importance of considering the missing data rate when selecting an imputation method. While all three methods performed well, previous research suggests that as missing data rates increase, MI and EM tend to outperform simpler approaches like CIM in maintaining factorial stability [39]. Mathematically, MI and EM rely on likelihood estimation principles, ensuring that missing data patterns do not distort parameter estimates. At the same time, CIM leverages the observed data distribution to approximate missing values efficiently. These methods remain effective across various missing rates because they preserve the covariance structure and factor loadings, which are critical for psychometric models, thereby ensuring the stability of the factor structure even as missing data rates increase. The consistency of the results with these studies reinforces the notion that selecting an appropriate imputation method depends on both the missing data rate and the complexity of the measurement model. Findings provide empirical support for the use of MI, EM, and CIM in maintaining the unidimensionality assumption, which is significant for valid factor analysis.

## Conclusion

The issue of missing data represents a major concern for data analysts, and it must be addressed seriously before analysis and decision-making. Data imputation methods are among the techniques used to handle missing data, making it ready for analysis by replacing missing values with the most probable ones, thus preserving the properties of the data to match and validate the assumptions of the statistical model that will be used for analysis. This paper discussed the impact of three data imputation methods: CIM, MI, and EM, across the assumption of MCAR, and different missing data rates, and their effect on the assumption of unidimensionality, using generated progressive data. After completing the imputation process with the three methods across missing data rates, the absolute differences between the imputed values and the original values were first calculated for the three methods and various missing rates to determine the level of distortion in the data post-imputation. Then, the assumption of unidimensionality was tested using five indicators: Cronbach's α, CITC, Eigenvalues, CTV, and communalities. The results also indicated that all the indicators used to test the unidimensionality assumption confirmed that the three imputation methods were effective in maintaining this assumption, regardless of the missing data rates. Furthermore, most of these indicators increased as the missing data rates increased. The study recommends conducting future research on the effectiveness of other imputation methods, and missing data mechanisms on the unidimensionality assumption or other assumptions of statistical models, with an emphasis on using real-world data when possible, this will help confirm the practical applicability of the findings and further explore the potential challenges that may arise in real-world data imputation.

## Supporting information

**Appendix 1. Corrected Item-Total Correlation (CITC) due to percentage of missingness**
(DOCX)

**Appendix 2. Eigenvalues of the first three components in detail for each imputation method**
(DOCX)

**Appendix 3. Communalities due to Percentages of Missingness through CIM, EM, and MI imputation methods**
(DOCX)

## Author contributions

**Conceptualization:** Ayman Omar Baniamer.

**Data curation:** Ayman Omar Baniamer.

**Formal analysis:** Ayman Omar Baniamer.

**Funding acquisition:** Ayman Omar Baniamer.

**Investigation:** Ayman Omar Baniamer.

**Methodology:** Ayman Omar Baniamer.

**Project administration:** Ayman Omar Baniamer.

**Resources:** Ayman Omar Baniamer.

**Software:** Ayman Omar Baniamer.

**Supervision:** Ayman Omar Baniamer.

**Validation:** Ayman Omar Baniamer.

**Visualization:** Ayman Omar Baniamer.

**Writing – original draft:** Ayman Omar Baniamer.

**Writing – review & editing:** Ayman Omar Baniamer.

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
