## [Decision Letter · Decision Letter 0]

21 Jan 2025

PONE-D-24-57472The Impact of Missing Data Rates and Imputation Methods on The Assumption of UnidimensionalityPLOS ONE

Dear Dr. Baniamer,

Thank you for submitting your manuscript to PLOS ONE. After careful consideration, we feel that it has merit but does not fully meet PLOS ONE’s publication criteria as it currently stands. Therefore, we invite you to submit a revised version of the manuscript that addresses the points raised during the review process.

We look forward to receiving your revised manuscript.

Kind regards,

Henri Tilga, PhD

Academic Editor

PLOS ONE

Journal requirements: When submitting your revision, we need you to address these additional requirements. 1. Please ensure that your manuscript meets PLOS ONE's style requirements, including those for file naming. The PLOS ONE style templates can be found at https://journals.plos.org/plosone/s/file?id=wjVg/PLOSOne_formatting_sample_main_body.pdf and https://journals.plos.org/plosone/s/file?id=ba62/PLOSOne_formatting_sample_title_authors_affiliations.pdf. 2. To meet PLOS ONE publication criteria (https://journals.plos.org/plosone/s/criteria-for-publication), manuscripts must ensure methods are described in sufficient detail for another researcher to reproduce the experiments. In your methods section, please provide additional information regarding the data source for the study. 3. Please provide a complete Data Availability Statement in the submission form, ensuring you include all necessary access information or a reason for why you are unable to make your data freely accessible. If your research concerns only data provided within your submission, please write "All data are in the manuscript and/or supporting information files" as your Data Availability Statement. 4. Please include captions for your Supporting Information files at the end of your manuscript, and update any in-text citations to match accordingly. Please see our Supporting Information guidelines for more information: http://journals.plos.org/plosone/s/supporting-information. 

Reviewers' comments:

Reviewer's Responses to Questions

**Comments to the Author**

1. Is the manuscript technically sound, and do the data support the conclusions?

Reviewer #1: Partly

Reviewer #2: Yes

Reviewer #3: Partly

2. Has the statistical analysis been performed appropriately and rigorously? 

Reviewer #1: Yes

Reviewer #2: Yes

Reviewer #3: No

3. Have the authors made all data underlying the findings in their manuscript fully available?

Reviewer #1: No

Reviewer #2: Yes

Reviewer #3: No

4. Is the manuscript presented in an intelligible fashion and written in standard English?

Reviewer #1: No

Reviewer #2: Yes

Reviewer #3: Yes

5. Review Comments to the Author

Reviewer #1: 1. The author need explain the assumption of unidimensionality, not every one know this, especially how it is related to your missing value imputation study.

2. The missing value imputation research has been studied by many researchers, different methods have been proposed, but the author only refer and consider a small amount of methods, in the past year, the generative methods have been used to impute missing values, the authors should include more recent publications, for example,

generative adversarial network based imputation methods:

Gain: Missing data imputation using generative adversarial nets, International conference on machine learning, 2018

A novel f-divergence based generative adversarial imputation method for scRNA-seq data analysis. Plos one 2023

Multivariate Time Series Imputation with Generative Adversarial Networks. In Proceedings of the NeurIPS, 2018

ImputeGAN: Generative adversarial network for multivariate time series imputation. Entropy 2023

VAE based method:

GP-VAE: Deep Probabilistic Time Series Imputation. In Proceedings of the AISTATS, 2020

Diffusion based method:

Csdi: Conditional score-based diffusion models for probabilistic time series imputation. Advances in Neural Information Processing Systems 2021,

There are a lot of new generative methods base imputation algorithm, you should cite more paper within 5 years.

3. The method is too simple, one suggestion, you should use a flowchart, or some pseudocode, to explain your method to make it more organized. The current method does not have too much novelty, very traditional methods.

4. The results section, the author use tables to show the comparison, I would suggest the authors use some figures to illustrate some of the results.

Reviewer #2: This is a study investigating the impact of missing data imputation methods on the assumption of unidimensionality. The study primarily focuses on comparing the performance of three imputation methods—CIM, EM, and MI—under varying levels of missing data proportions, and employs multiple indicators to evaluate unidimensionality. However, the research has notable limitations, such as a lack of theoretical justification for the choice of imputation methods and the exclusive use of simulated data without incorporating real-world datasets.

Key Issues

1. Insufficient Theoretical Justification for Imputation Method Selection

The paper does not provide a sufficient theoretical basis for selecting CIM, EM, and MI as the focal imputation methods. To improve the study’s rigor, a detailed discussion should be added to justify why these methods were chosen over others.

2. Reliance on Simulated Data Without Validation Using Real-World Data

The analysis is based solely on simulated data, which may undermine the external validity of the findings. It is suggested that future studies incorporate real-world data to validate and extend the conclusions.

3. Ambiguities in Formula Descriptions and Symbol Definitions

The study contains issues with the descriptions of formulas, including ambiguous or flawed symbol definitions. This raises questions about the accuracy of the analysis. If there are errors in the definitions, it is unclear how the study was able to proceed with its investigations. A thorough clarification and correction of the formulas and their definitions are strongly recommended.

4. Lack of Distinction from Existing Literature

The study's topic has already been extensively explored in prior research. However, the authors fail to articulate the unique contribution or novel aspects of their work. For example, several studies, such as the following, have already addressed similar topic

Newman, D.A. (2003). Longitudinal Modeling with Randomly and Systematically Missing Data: A Simulation of Ad Hoc, Maximum Likelihood, and Multiple Imputation Techniques. Organizational Research Methods, 6, 328 - 362.

Chukwu, A.U., Ezichi, O.N., & DikeA., O. (2015). On Comparison of Some Imputation Techniques in Multivariate Data Analysis. Mathematical theory and modeling, 5, 95-110.

Di̇nçsoy, L.B., & Kelecioğlu, H. (2022). INVESTIGATION OF THE EFFECT OF MISSING DATA ON DIFFERANTIAL ITEM FUNCTIONING IN MIXED TYPE TESTS. Eğitimde ve Psikolojide Ölçme ve Değerlendirme Dergisi.

Cheung, M.W. (2007). Comparison of Methods of Handling Missing Time-Invariant Covariates in Latent Growth Models Under the Assumption of Missing Completely at Random. Organizational Research Methods, 10, 609 - 634.

5. Theoretical Justification for the Selection of CIM, EM, and MI

As mentioned earlier, there is a lack of theoretical discussion regarding the choice of CIM, EM, and MI. Additional details should be provided to explain why these specific methods were selected.

Secondary Issues

1. Inadequate Details on Unidimensionality Indicators in the Abstract

The abstract lacks a clear explanation of the specific unidimensionality indicators employed in the study. This information should be added to provide a more comprehensive overview of the methodology.

2.. Superficial Discussion of the Importance of Unidimensionality in the Introduction

The introduction provides only a cursory explanation of the importance of the unidimensionality assumption. A deeper discussion of its theoretical and practical significance is recommended.

3. Unclear Table Numbering and Redundant Content

Some tables in the results section are poorly numbered, and there is redundancy in the table content. A clearer and more concise presentation of the tables would improve the overall readability.

4. Insufficient Analysis of Study Limitations in the Discussion Section

The discussion section does not provide an in-depth analysis of the study's limitations. For example, the lack of real-world data and potential biases in the selection of missing data proportions are not sufficiently addressed. Expanding on these aspects would enhance the comprehensiveness of the discussion.

Reviewer #3: The paper examined the impact of missing data rates and imputation methods on fulfilling the assumption of unidimensionality, a core assumption supporting many statistical models. The importance/significance of conducting such study is not clearly highlighted in the paper. There are several similar studies that have been conducted, for instance the reference [19] has compared 6 methods for handling missing data, [49] compared 12 imputation techniques, and [30] in which 4 imputation techniques have been compared; comparing and highlighting their differences in a section (related work) would perhaps justify the motivation of conducting the study.

A clear formal definition of unidimensionality should be given. A section on motivation of the study should be included to justify the significance of conducting the analysis.

In the paper, the authors have examined three methods for imputing missing values, namely: multiple imputation (MI), expectation maximization algorithm (EM), and corrected item mean (CIM). Authors should deliberate on all possible methods for imputing missing values and justify the reasons for focusing only on the above three methods. Also, there are three mechanisms of missing data as defined in [17] that are Missing Completely At Random (MCAR), Missing At Random (MAR), and Missing Not at Random. While the study has assumed MCAR with missing rates of 1% … 50%. Justification on the selection of mechanism, i.e. MCAR, and the range of missing values is not clear. Why 50% is set as the highest missing rate? Also, the selection of 5000 examinees and a test of 50 items is not obvious. Moreover, performing the analysis on real dataset might result in more profound results.

The discussion section is not well written as it merely reports the findings with regard to Cronbach’s α, CITC, Eigenvalues, CTV, and communalities which have been reported in the earlier sections. Moreover, the findings and discussions are based on the selected imputation techniques. It is not comprehensive as there might be other imputation techniques that would show better results.

6. PLOS authors have the option to publish the peer review history of their article (what does this mean? ). If published, this will include your full peer review and any attached files.

**Do you want your identity to be public for this peer review?** For information about this choice, including consent withdrawal, please see our Privacy Policy .

Reviewer #1: No

Reviewer #2: No

Reviewer #3: **Yes: ** Hamidah Ibrahim

---

## [Author Response · Author response to Decision Letter 1]

26 Feb 2025

Response to Journal and Reviewers comment

Response to the journal requirements

We appreciate your careful review of our manuscript and thank you for the opportunity to submit our revised manuscript and the point-by-point responses to the journal requirements. We hope that we have satisfactorily addressed all issues raised by the journal requirements.

Comment

Comment 1. Please ensure that your manuscript meets PLOS ONE's style requirements, including those for file naming. The PLOS ONE style templates can be found at

Response 1: Done

Comment 2. To meet PLOS ONE publication criteria (https://journals.plos.org/plosone/s/criteria-for-publication), manuscripts must ensure methods are described in sufficient detail for another researcher to reproduce the experiments. In your methods section, please provide additional information regarding the data source for the study.

Response 2: As requested, the information regarding data were illustrated in the method section. Please see our revised manuscript. Done

Comment 3. Please provide a complete Data Availability Statement in the submission form, ensuring you include all necessary access information or a reason for why you are unable to make your data freely accessible. If your research concerns only data provided within your submission, please write "All data are in the manuscript and/or supporting information files" as your Data Availability Statement.

Response 3: The data underlying the results presented in the study are available on the following link: 10.6084/m9.figshare.28354868

Comment 4: Please include captions for your Supporting Information files at the end of your manuscript, and update any in-text citations to match accordingly. Please see our Supporting Information guidelines for more information: http://journals.plos.org/plosone/s/supporting-information.

Response 4: Captions of my appendixes 1, 2 & 3 are included at the end of the manuscript.

Appendix (1): Corrected Item-Total Correlation (CITC) DUE TO PERCENTAGE OF MISSINGNESS

Appendix (2): Eigenvalues of the first three components in detail for each imputation method

Appendix (3): Communalities due to Percentages of Missingness through CIM, EM, and MI imputation methods

Dear Editor,

We would like to thank you and the reviewers for their valuable comments and feedback. Point-by-point responses to reviewers are listed below.

Reviewer comments

Reviewer #1:

Comment 1: The author need explain the assumption of unidimensionality, not every one know this, especially how it is related to your missing value imputation study.

Response 1: Thank you for your comments. We clarified the assumption of unidimensionality in our missing value imputation study. Please see our revised manuscript.

Comment 2: author only refer and consider a small amount of methods, in the past year, the generative methods have been used to impute missing values, the authors should include more recent publications, for example, generative adversarial network based imputation methods:

Gain: Missing data imputation using generative adversarial nets, International conference on machine learning, 2018

A novel f-divergence based generative adversarial imputation method for scRNA-seq data analysis. Plos one 2023

Multivariate Time Series Imputation with Generative Adversarial Networks. In Proceedings of the NeurIPS, 2018

ImputeGAN: Generative adversarial network for multivariate time series imputation. Entropy 2023

VAE based method:

GP-VAE: Deep Probabilistic Time Series Imputation. In Proceedings of the AISTATS, 2020

Diffusion based method:

Csdi: Conditional score-based diffusion models for probabilistic time series imputation. Advances in Neural Information Processing Systems 2021,

There are a lot of new generative methods base imputation algorithm, you should cite more paper within 5 years.

Response 2: Thank you for pointing out the importance of considering recent advancements in imputation methods, particularly generative approaches. We acknowledge that our current review may not fully capture the latest developments in this field. We incorporated these studies and others from the past 5 years into our review to provide a more comprehensive overview of recent advancements in imputation methods.

Comment 3: The method is too simple, one suggestion, you should use a flowchart, or some pseudocode, to explain your method to make it more organized. The current method does not have too much novelty, very traditional methods.

Response 3: To address your suggestion, we revised our manuscript to include a flowchart and pseudocode to better illustrate our method and improve its clarity. Please see our revised manuscript containing flowchart.

Comment 4: The results section, the author use tables to show the comparison, I would suggest the authors use some figures to illustrate some of the results.

Response 4: Thank you for the suggestion. I believe the Tables sufficiently illustrate the comparisons, making additional figures unnecessary.

Reviewer #2: This is a study investigating the impact of missing data imputation methods on the assumption of unidimensionality. The study primarily focuses on comparing the performance of three imputation methods—CIM, EM, and MI—under varying levels of missing data proportions, and employs multiple indicators to evaluate unidimensionality. However, the research has notable limitations, such as a lack of theoretical justification for the choice of imputation methods and the exclusive use of simulated data without incorporating real-world datasets.

Key Issues

Comment 1: Insufficient Theoretical Justification for Imputation Method Selection. The paper does not provide a sufficient theoretical basis for selecting CIM, EM, and MI as the focal imputation methods. To improve the study’s rigor, a detailed discussion should be added to justify why these methods were chosen over others.

Response 1: To address this concern, we added a new section to our manuscript that provides a comprehensive review of the theoretical underpinnings of CIM, EM, and MI. We discussed the strengths and limitations of each method, as well as their suitability for our specific research context.

Comment 2: Reliance on Simulated Data Without Validation Using Real-World Data. The analysis is based solely on simulated data, which may undermine the external validity of the findings. It is suggested that future studies incorporate real-world data to validate and extend the conclusions.

Response 2: done, I appreciate the reviewer’s concern regarding the reliance on simulated data and its potential impact on the external validity of the findings. I want to clarify this point as follows:

o The decision to employ simulated data was driven by the need to maintain complete control over the missing data mechanism, the underlying unidimensional structure, and other key parameters. That allowed the researcher to systematically assess the impact of different imputation methods under well-defined conditions (MCAR) and isolate the imputation techniques' effects the unidimensionality assumption. This baseline facilitates clear comparisons and enhances the interpretability of the findings, which can be later built upon using more complex, real-world datasets.

o It's clear that findings based solely on simulated data may not fully capture all the nuances present in real-world scenarios. To address this limitation, the researcher clearly stated in the manuscript that real-world validation is a necessary next step. This will help confirm the practical applicability of the findings and further explore the potential challenges that may arise in real-world data imputation.

Comment 3: Ambiguities in Formula Descriptions and Symbol Definitions. The study contains issues with the descriptions of formulas, including ambiguous or flawed symbol definitions. This raises questions about the accuracy of the analysis. If there are errors in the definitions, it is unclear how the study was able to proceed with its investigations. A thorough clarification and correction of the formulas and their definitions are strongly recommended.

Response 3: We take these concerns seriously and will thoroughly review and correct our formulas and symbol definitions to ensure accuracy and clarity. We will provide explicit definitions for all symbols and ensure that our formulas are correctly described and consistent throughout the manuscript.

All manipulations were reviewed to make sure of their accuracy, and all formulas were corrected, and to be sure all manipulations were uploaded on the links: 10.6084/m9.figshare.28387778

10.6084/m9.figshare.28462568

10.6084/m9.figshare.28462607

Comment 4: Lack of Distinction from Existing Literature. The study's topic has already been extensively explored in prior research. However, the authors fail to articulate the unique contribution or novel aspects of their work. For example, several studies, such as the following, have already addressed similar topic

Newman, D.A. (2003). Longitudinal Modeling with Randomly and Systematically Missing Data: A Simulation of Ad Hoc, Maximum Likelihood, and Multiple Imputation Techniques. Organizational Research Methods, 6, 328 - 362.

Chukwu, A.U., Ezichi, O.N., & DikeA., O. (2015). On Comparison of Some Imputation Techniques in Multivariate Data Analysis. Mathematical theory and modeling, 5, 95-110.

Di̇nçsoy, L.B., & Kelecioğlu, H. (2022). INVESTIGATION OF THE EFFECT OF MISSING DATA ON DIFFERANTIAL ITEM FUNCTIONING IN MIXED TYPE TESTS. Eğitimde ve Psikolojide Ölçme ve Değerlendirme Dergisi.

Cheung, M.W. (2007). Comparison of Methods of Handling Missing Time-Invariant Covariates in Latent Growth Models Under the Assumption of Missing Completely at Random. Organizational Research Methods, 10, 609 - 634.

Response 4: We mentioned in our introduction, our study focuses on the specific topic of evaluating the effect of different imputation methods on the assumption of unidimensionality. To our knowledge, no prior study has explicitly addressed this research question.

Comment 5: Theoretical Justification for the Selection of CIM, EM, and MI. As mentioned earlier, there is a lack of theoretical discussion regarding the choice of CIM, EM, and MI. Additional details should be provided to explain why these specific methods were selected.

Response 5: Added. Please see our revised manuscript.

Secondary Issues

Comment 6:

1. Inadequate Details on Unidimensionality Indicators in the Abstract. The abstract lacks a clear explanation of the specific unidimensionality indicators employed in the study. This information should be added to provide a more comprehensive overview of the methodology.

Response 6: We revised the abstract to include a clear explanation of the specific unidimensionality indicators used in the study.

Comment 7:

2.. Superficial Discussion of the Importance of Unidimensionality in the Introduction

The introduction provides only a cursory explanation of the importance of the unidimensionality assumption. A deeper discussion of its theoretical and practical significance is recommended.

Response 7: Done

Comment 8:

3. Unclear Table Numbering and Redundant Content

Some tables in the results section are poorly numbered, and there is redundancy in the table content. A clearer and more concise presentation of the tables would improve the overall readability.

Response 8: done, All tables have been renumbered correctly. However, regarding the table contents, each table has been included for a specific purpose, and there is no duplication in the data, and I think they are clear.

Comment 9:

4. Insufficient Analysis of Study Limitations in the Discussion Section

The discussion section does not provide an in-depth analysis of the study's limitations. For example, the lack of real-world data and potential biases in the selection of missing data proportions are not sufficiently addressed. Expanding on these aspects would enhance the comprehensiveness of the discussion.

Response 9: done, all study limitations were mentioned in the discussion and conclusion

Reviewer #3:

Comment 1: The paper examined the impact of missing data rates and imputation methods on fulfilling the assumption of unidimensionality, a core assumption supporting many statistical models. The importance/significance of conducting such study is not clearly highlighted in the paper. There are several similar studies that have been conducted, for instance the reference [19] has compared 6 methods for handling missing data, [49] compared 12 imputation techniques, and [30] in which 4 imputation techniques have been compared; comparing and highlighting their differences in a section (related work) would perhaps justify the motivation of conducting the study.

Response 1: While numerous studies have compared various imputation methods across different fields, our study uniquely investigates the impact of these methods on the assumption of unidimensionality, as outlined in the introduction.

Comment 2: A clear formal definition of unidimensionality should be given. A section on motivation of the study should be included to justify the significance of conducting the analysis.

Response 2: We added a formal definition of unidimensionality to the revised manuscript. Comment 3: In the paper, the authors have examined three methods for imputing missing values, namely: multiple imputation (MI), expectation maximization algorithm (EM), and corrected item mean (CIM). Authors should deliberate on all possible methods for imputing missing values and justify the reasons for focusing only on the above three methods. Also, there are three mechanisms of missing data as defined in [17] that are Missing Completely At Random (MCAR), Missing At Random (MAR), and Missing Not at Random. While the study has assumed MCAR with missing rates of 1% … 50%. Justification on the selection of mechanism, i.e. MCAR, and the range of missing values is not clear. Why 50% is set as the highest missing rate? Also, the selection of 5000 examinees and a test of 50 items is not obvious. Moreover, performing the analysis on real dataset might result in more profound results.

Response 3: Done, as your suggestion.

Comment 4: The discussion section is not well written as it merely reports the findings with regard to Cronbach’s α, CITC, Eigenvalues, CTV, and communalities which have been reported in the earlier sections. Moreover, the findings and discussions are based on the selected imputation techniques. It is not comprehensive as there might be other imputation techniques that would show better results.

Response 4: We revised the discussion section to provide a more comprehensive interpretation of the results, beyond just summarizing the findings. We will also discuss the implications of our results, including the potential limitations of the selected imputation techniques and the possibility of other techniques yielding better results.

We hope now that our revised manuscript is acceptable for publication.

---

## [Decision Letter · Decision Letter 1]

5 Mar 2025

The Impact of Missing Data Rates and Imputation Methods on The Assumption of Unidimensionality

PONE-D-24-57472R1

Dear Dr. Baniamer,

We’re pleased to inform you that your manuscript has been judged scientifically suitable for publication and will be formally accepted for publication once it meets all outstanding technical requirements.

Kind regards,

Henri Tilga, PhD

Academic Editor

PLOS ONE

Additional Editor Comments (optional):

Reviewers' comments:

Reviewer's Responses to Questions

**Comments to the Author**

1. If the authors have adequately addressed your comments raised in a previous round of review and you feel that this manuscript is now acceptable for publication, you may indicate that here to bypass the “Comments to the Author” section, enter your conflict of interest statement in the “Confidential to Editor” section, and submit your "Accept" recommendation.

Reviewer #1: All comments have been addressed

Reviewer #2: All comments have been addressed

2. Is the manuscript technically sound, and do the data support the conclusions?

Reviewer #1: Yes

Reviewer #2: Yes

3. Has the statistical analysis been performed appropriately and rigorously? 

Reviewer #1: Yes

Reviewer #2: Yes

4. Have the authors made all data underlying the findings in their manuscript fully available?

Reviewer #1: Yes

Reviewer #2: Yes

5. Is the manuscript presented in an intelligible fashion and written in standard English?

Reviewer #1: Yes

Reviewer #2: Yes

6. Review Comments to the Author

Reviewer #1: (No Response)

Reviewer #2: In the revised manuscript, it is evident that the author has made every effort to address all the raised concerns.

7. PLOS authors have the option to publish the peer review history of their article (what does this mean? ). If published, this will include your full peer review and any attached files.

**Do you want your identity to be public for this peer review?** For information about this choice, including consent withdrawal, please see our Privacy Policy .

Reviewer #1: No

Reviewer #2: No

---

## [Editor Report · Acceptance letter]

PONE-D-24-57472R1

PLOS ONE

Dear Dr. Baniamer,

I'm pleased to inform you that your manuscript has been deemed suitable for publication in PLOS ONE. Congratulations! Your manuscript is now being handed over to our production team.

Kind regards,

on behalf of

Dr. Henri Tilga

Academic Editor

PLOS ONE